# Movable Antenna-Enabled RIS-Assisted Simultaneous Wireless Information and Power Transfer Systems

**DOI:** 10.3390/s25237402

**Published:** 2025-12-04

**Authors:** Dun Feng, Xuan Zhang, Xiaofan Yu, Xin Wang, Xiaoye Shi

**Affiliations:** 1College of Telecommunications and Information Engineering, Nanjing University of Posts and Telecommunications, Nanjing 210003, China; fengdun.js@chinatelecom.cn (D.F.); 20250071@njupt.edu.cn (X.Z.); shixy187@njupt.edu.cn (X.S.); 2Jiangsu Branch, China Telecom Corporation Ltd., Nanjing 210008, China; wx1.js@chinatelecom.cn

**Keywords:** Internet of Things (IoT), Movable Antenna (MA), Reconfigurable Intelligent Surfaces (RIS), Simultaneous Wireless Information and Power Transfer (SWIPT)

## Abstract

The integration of movable antenna (MA) and reconfigurable intelligent surfaces (RIS) offers promising potential for enhancing simultaneous wireless information and power transfer (SWIPT) systems. In this paper, we investigate a novel MA-enabled RIS-assisted SWIPT framework, where both RIS and MA are jointly exploited to provide additional spatial degrees of freedom and reconfigurable propagation channels. Then, we formulate an energy harvesting maximization problem under communication reliability constraints by jointly optimizing the base station beamforming, RIS phase shifts, and MA positions. To tackle the proposed non-convexity problem, an efficient alternating optimization (AO) algorithm is developed, which is based on successive convex approximation (SCA) and second-order Taylor expansion. The obtained simulation outcomes reveal that incorporating MA into RIS-assisted SWIPT systems leads to notable performance gains over both conventional RIS schemes and fixed-antenna benchmarks.

## 1. Introduction

In recent years, the Internet of Things (IoT) has become increasingly intertwined with various aspects of daily life, particularly in the development of smart agriculture, smart homes, and intelligent transportation systems [1]. With the explosive growth in the number of IoT nodes, efficient and reliable methods for both energy supply and information transmission have become critical challenges. Moreover, battery-powered devices often suffer from limited operational lifespans, and it is difficult or even impractical to replace or recharge these batteries in certain deployment scenarios [2]. Because radio-frequency signals inherently carry both energy and information, simultaneous wireless information and power transfer (SWIPT) has become an attractive and eco-friendly approach to overcome these limitations [3,4,5]. Over the past decades, beam pattern synthesis has undergone development, providing both analytical foundations for emerging SWIPT systems. Classical works such as [6,7] for narrow-beam and sidelobe-controlled array structures. The practical implementation of SWIPT systems fundamentally relies on advanced beamforming and array synthesis techniques [8,9,10,11]. By shaping energy-efficient beams with spatial selectivity, beamforming enables targeted delivery of both information and energy, thereby serving as the backbone of SWIPT-enabled architectures.

Reconfigurable intelligent surfaces (RIS) have become a key technology for future wireless networks, thanks to its advantages of low energy consumption, low latency, and high deployment feasibility [12,13]. By dynamically adjusting the amplitude and phase of incident signals, RIS can intelligently reconfigure the wireless propagation environment, thereby offering significant benefits in terms of energy efficiency, coverage extension, and spectrum utilization [14]. Particularly in complex and highly dynamic scenarios, RIS is envisioned to effectively compensate for coverage blind spots that the conventional base station (BS) cannot address, thus ensuring ubiquitous and reliable connectivity [15,16]. Due to its ability to enhance energy-harvesting (EH) and information-transmission capabilities, the integration of RIS with SWIPT has recently garnered significant attention [17,18,19]. RIS-assisted SWIPT not only improves the efficiency of wireless power transfer but also guarantees reliable information delivery, thereby providing practical and sustainable solutions for emerging applications such as the IoT, smart agriculture, and smart cities.

In contrast to RIS, which provides additional transmission paths by constructing virtual line-of-sight links, the recently emerging fluid antenna system (FAS) [20] and movable antenna (MA) [21] technologies have gained considerable attention due to their capability of dynamically adjusting the position, orientation, and configuration of transceiver antennas. When the reflected beam of the RIS lacks sufficient focus or alignment, the MA can provide fine-grained spatial adaptation to compensate for beamforming imperfections, thereby enhancing overall beam directivity and performance. Unlike conventional fixed-position antenna (FPA), which cannot exploit the spatial dynamics of wireless channels, MAs can physically move within a predefined region to reshape the channel characteristics, thereby enabling effective interference suppression, signal enhancement, and coverage optimization. Specifically, MAs are typically connected to radio-frequency chains via flexible cables and controlled by compact actuators to achieve continuous movement within a spatial region [22], thus overcoming the inherent limitations of FPA. Leveraging such flexibility, MA can fully exploit the spatial variability of wireless channels, offering stronger adaptability and resilience while improving signal reception quality, transmission efficiency, and energy efficiency [23]. Notably, the spatial degrees of freedom (DoF) and energy efficiency benefits enabled by MAs also open up new opportunities for their integration with SWIPT.

### 1.1. Related Works

In recent years, combining SWIPT with RIS and MA to enhance EH and reliable communication performance has garnered significant attention. This paper provides a brief review and discussion of the current state of the art.

#### 1.1.1. RIS-Assisted SWIPT

Research on RIS-assisted SWIPT systems has become extensive, demonstrating that RIS can significantly enhance information transmission while improving EH performance. For instance, the paper [24] investigated an RIS-assisted SWIPT system where BS communicates with multiple information receivers (IRs) while simultaneously satisfying the EH demands of energy receivers (ERs). In combination with other communication techniques, the authors in [25] proposed an RIS-assisted SWIPT with non-orthogonal multiple access (NOMA) architecture, which improves both the transmission performance of NOMA users and the energy transfer efficiency of SWIPT. Moreover, the researchers in [26] explored the integration of RIS into SWIPT-enabled mobile edge computing networks, highlighting its potential to enhance spectral and energy efficiency. Beyond the conventional linear EH model, some studies have incorporated more realistic nonlinear EH models. In [27], a large-scale RIS-aided SWIPT system was examined, and a two-stage scalable optimization strategy—comprising offline design and online adaptation—was introduced. Similarly, the paper [28] investigated an RIS-assisted orthogonal frequency division multiplexing SWIPT system under a nonlinear EH model, thereby better capturing practical hardware characteristics.

#### 1.1.2. MA-Assisted SWIPT

MA/FAS can dynamically reconfigure the wireless propagation environment by flexibly adjusting the antenna position and orientation, thereby introducing additional spatial DoF and achieving performance gains. Such capability also holds great potential for SWIPT systems. For instance, the researchers in [29] proposed the FAS-assisted SWIPT framework, where the channel reconfiguration ability of FAS was leveraged to effectively improve both EH and communication performance. In addition, the authors in [30] investigated the MA-assisted SWIPT secure-transmission scheme, which enhanced EH efficiency while ensuring physical layer security. Although these studies demonstrate the promising potential of MA/FAS in SWIPT, compared with the extensive literature on RIS-assisted SWIPT, research in this direction is still in its infancy and requires further exploration.

### 1.2. Motivation and Contributions

Overall, existing studies have demonstrated the promising potential of MA in enhancing SWIPT systems. However, current research on this topic remains limited, particularly regarding the joint utilization of MA and RIS. To the best of our knowledge, this is the first study that investigates the integration of RIS and MA in SWIPT systems. The key contributions can be summarized as follows:In this paper, we establish a novel RIS-assisted SWIPT system model empowered by MA, which systematically characterizes the joint effect of RIS and MA in providing spatial DoF and channel reconfiguration, while addressing both information transmission and EH requirements. Subsequently, an optimization model is formulated to enhance energy harvesting, under the condition that reliable communication quality is preserved.To address the strongly non-convex joint optimization of transmit beamforming, RIS reflection coefficients, and MA positions, we devise an efficient alternating optimization (AO) framework that employs techniques such as successive convex approximation (SCA) and the second-order Taylor expansion.Simulations are carried out to assess the effectiveness of the MA-enabled RIS-assisted SWIPT design. The results confirm that this scheme significantly enhances energy-harvesting performance while maintaining reliable communication, surpassing conventional RIS-assisted SWIPT and fixed-position antenna benchmarks.

### 1.3. Notations

We use |·| to denote scalar magnitude. ${\left(\cdot \right)}^{T}$ and ${\left(\cdot \right)}^{H}$ correspond to the transpose and Hermitian transpose. The notation $C^{a\times b}$ specifies the complex matrices with size a×b, while E[·] represents the expectation. A random variable distributed according to a complex Gaussian law with mean *a* and variance *b* is expressed as x∼CN(a,b); the operator ∠(·) gives the phase. Table 1 summarizes the abbreviations frequently used in this paper.

## 2. System Model and Problem Formulation

We consider a downlink RIS-assisted SWIPT system in Figure 1, where the MA is deployed at the receiver side, i.e., at the locations of both the IoT node and the IRs. The system comprises *K* IRs and a single IoT node (Although the present work examines a single IoT node, the adopted modeling methodology is flexible and can be extended to configurations with multiple IoT nodes. Furthermore, the algorithm can be adapted to multi IoT nodes with only minor adjustments.). The IRs are solely responsible for decoding information. In contrast, the IoT node serves a dual role in that it decodes the transmitted signal and simultaneously harvests energy from the received signals for self-sustainable operation; thus, it acts as both an IR and an ER. Due to the presence of a significant obstruction between the BS and the IRs and IoT node, an RIS consisting of *M* passive elements is deployed to establish indirect communication links. Moreover, the BS is equipped with an array of *N* transmit antennas. Each IR user and the IoT node is equipped with an MA to enhance the received signal strength via position optimization. We denote the antenna location of the receiver as $s_{i}=\left[x_{i},y_{i}\right]\in A_{i},i\in \left\{k,e\right\}$, with $k\in \left\{1,2,\ldots ,K\right\}$ representing the IR and *e* representing the IoT node. The MA is constrained within a predefined rectangular region defined as $A_{i}=\left[-\frac{D_{i}}{2},\frac{D_{i}}{2}\right]\times \left[-\frac{D_{i}}{2},\frac{D_{i}}{2}\right]$, which specifies the allowable movement area for each user-side MA. For convenience, we define $M=\left\{1,2,\ldots ,M\right\}$, and $K=\left\{1,2,\ldots ,K\right\}$.

### 2.1. Channel Model

To characterize the spatial wireless channels in the considered system, we adopt the field response channel model [23] (This paper aims to provide a theoretical performance benchmark and a proof-of-concept reference. We will consider extending the analysis to more generalized channel conditions in our future work.). Taking the RIS-to-user link as an example, we consider a multi-path propagation environment. For the *i*-th receiver, let $\varphi_{i,l}^{r}$ and $\phi_{i,l}^{r}$ denote the elevation and azimuth angles of arrival for the $l_{i}$-th path of the RIS-user link, respectively, where $i\in \left\{k,e\right\}$, k∈K, $1\leq l_{i}\leq L_{i}^{r}$, and $L_{i}^{r}$ is the total number of receive paths. These angles capture the spatial directionality of each reflected component at the user side. Based on the field-effect model [23], we present the following channel model; the phase difference in signal propagation between transmission path $l_{i}$ and the reference point is expressed as(1)$$\rho_{i,l_{i}}^{r}\left(s_{i}\right)=x_{i}\sin\varphi_{i,l_{i}}^{r}\cos\phi_{i,l_{i}}^{r}+y_{i}\cos\varphi_{i,l_{i}}^{r}.$$

Subsequently, the field response vector (FRV) received by MA at receive node *i* can be given by(2)$$g_{si}\left(s_{i}\right)={\left[e^{j\frac{2\pi }{\lambda }\rho_{i,1}^{r}\left(s_{i}\right)},e^{j\frac{2\pi }{\lambda }\rho_{i,2}^{r}\left(s_{i}\right)},\ldots ,e^{j\frac{2\pi }{\lambda }\rho_{i,L_{i}^{r}}^{r}\left(s_{i}\right)}\right]}^{T},i\in \left\{k,e\right\},k\in K,$$
where λ represents the wavelength. Furthermore, we set the elevation angle and azimuth angle of the *p*-th path in the RIS transmit transmission links to be $\varphi_{s,p}^{t}$ and $\phi_{s,p}^{t}$, respectively, where $1\leq p\leq L_{s}^{t}$ and $L_{s}^{t}$ is the total number of transmit paths. The signal propagation phase difference between the *p*-th path of the *m*-th element and the reference position is given as(3)$$\rho_{s,p}^{t}\left(c_{m}\right)=x_{m}\sin\varphi_{s,p}^{t}\cos\phi_{s,p}^{t}+y_{m}\cos\varphi_{s,p}^{t}.$$

Similarly, the transmit FRV for the *m*-th element is(4)$$f\left(c_{m}\right)={\left[e^{j\frac{2\pi }{\lambda }\rho_{s,1}^{t}\left(c_{m}\right)},e^{j\frac{2\pi }{\lambda }\rho_{s,2}^{t}\left(c_{m}\right)},\ldots ,e^{j\frac{2\pi }{\lambda }\rho_{s,L_{s}^{t}}^{t}\left(c_{m}\right)}\right]}^{T}.$$

Thus, we can derive the transmit field response matrix (FRM) for RIS as(5)$$F_{si}=\left[f\left(c_{1}\right),f\left(c_{2}\right),\ldots ,f\left(c_{M}\right)\right].$$

Based on the above, the channel between RIS and receive node *i* is(6)$$h_{s,i}=F_{si}^{H}\Sigma_{si}g_{si}\left(s_{i}\right),i\in \left\{k,e\right\},k\in K,$$
where $\Sigma_{si}\in C^{L_{s}^{t}\times L_{s}^{r}}$ denotes the path response matrix. Based on the above channel formation process, we can derive the channel from the BS to the RIS as(7)$$H=G_{br}\Sigma_{br}F_{br}^{H}\in C^{M\times N},$$
where $G_{br}\in C^{M\times L_{b}^{r}}$ and $F_{br}\in C^{N\times L_{b}^{t}}$ represent the receive and transmit FRM, respectively, and $\Sigma_{br}\in C^{L_{b}^{r}\times L_{b}^{t}}$ denotes the path response matrix [23].

### 2.2. Signal Model

To simultaneously support information transmission to all IRs and ensure sufficient power transfer to the IoT node, the transmit signal at the BS is modeled as(8)$$x=\sum_{k=1}^{K}w_{k}c_{k}+w_{e}c_{e}.$$
where $w_{k}\in C^{N\times 1}$ denotes the beamforming vector for the IR *k*, and $c_{k}$ represents the corresponding data symbol with $E\left\{{\left|c_{k}\right|}^{2}\right\}=1$. Similarly, $w_{e}\in C^{N\times 1}$ and $c_{e}$ with $E\left\{{\left|c_{e}\right|}^{2}\right\}=1$ are the beamforming vector and data symbol intended for the IoT node. We assume that all data symbols are mutually independent and uncorrelated. Accordingly, the received baseband signal at the node *i* is given by(9)$$y_{i}=\sum_{k=1}^{K}h_{s,i}^{H}\Phi Hw_{k}c_{k}+h_{s,i}^{H}\Phi Hw_{e}c_{e}+n_{i},i\in \left\{k,e\right\},k\in K,$$
where $n_{i}∼\mathrm{CN}\left(0,\sigma_{i}^{2}\right)$ denotes the additive white Gaussian noise (AWGN) at receiver *i*. The matrix $\Phi =diag\left(\theta_{1},\theta_{2},\ldots ,\theta_{M}\right)$ represents the reflection coefficients of the RIS, where each element satisfies $\left|\theta_{m}\right|=1$. According to [2,30], the IoT node adopts a power splitting architecture to simultaneously perform information decoding and EH. Specifically, the received signal is divided into two streams by a power splitter; a splitting ratio $\beta \in \left[0,1\right]$ is allocated for information decoding, while the remaining 1−β is used for EH. Assuming that the noise introduced during the splitting process is negligible, the signal used for ID at the IoT node can be modeled as(10)$$y_{e}^{\mathrm{ID}}=\sqrt{\beta }y_{e}+n_{s}^{2}.$$
where $n_{s}∼\mathrm{CN}\left(0,\sigma_{s}^{2}\right)$ is the AWGN at the IoT receiver.

Based on the above system model, the signal-to-interference-plus-noise ratio (SINR) at the IR *k* and the IoT node can be, respectively, expressed as(11)$$\gamma_{k}=\frac{{\left|h_{s,k}^{H}\Phi Hw_{k}\right|}^{2}}{\sum_{i\neq k}^{K}{\left|h_{s,k}^{H}\Phi Hw_{i}\right|}^{2}+{\left|h_{s,k}^{H}\Phi Hw_{e}\right|}^{2}+\sigma_{k}^{2}},k\in K,$$
and(12)$$\gamma_{e}=\frac{{\left|h_{s,e}^{H}\Phi Hw_{e}\right|}^{2}}{\sum_{k=1}^{K}{\left|h_{s,e}^{H}\Phi Hw_{k}\right|}^{2}+\sigma_{e,d}^{2}},$$
where $\sigma_{e,d}^{2}=\sigma_{e}^{2}+\frac{\sigma_{s}^{2}}{\beta }$ denotes the effective noise power at the IoT node during ID, accounting for both ambient AWGN and the impact of power splitting. Furthermore, under a linear EH model [30], the amount of power harvested at the IoT node is given by(13)$$E_{h}=\eta \left(1-\beta \right)\left(\sum_{k=1}^{K}{\left|h_{s,e}^{H}\Phi Hw_{k}\right|}^{2}+{\left|h_{s,e}^{H}\Phi Hw_{e}\right|}^{2}\right),$$
where $\eta \in \left[0,1\right]$ denotes the EH efficiency coefficient.

### 2.3. Problem Formulation

To enable simultaneous information transmission and EH in the considered MA-enabled RIS-assisted system, we design an optimization framework aimed at maximizing the energy harvested by the IoT node. This objective is pursued through joint optimization of the BS beamforming vectors, RIS reflection coefficients, and MA positions at the receivers. Meanwhile, to ensure reliable communication, minimum SINR constraints are imposed on both the IR users and the IoT node. Additionally, constraints on total transmit power, unit-modulus RIS phase shifts, and feasible antenna positions are imposed. The resulting problem is formulated as(14)$$\begin{array}{cc} \; & P1:\max_{{\left\{w_{k},s_{k}\right\}}_{k=1}^{K},w_{e},\Phi ,s_{e}}E_{h} \end{array}$$(14a)$$\begin{array}{cc} \; & \;\;\;\;\;\;\;\;\;s.t.\;\gamma_{k}\geq \gamma_{k,\mathrm{th}},k\in K, \end{array}$$(14b)$$\begin{array}{cc} \; & \;\;\;\;\;\;\;\;\;\;\;\;\;\;\;\gamma_{e}\geq \gamma_{e,\mathrm{th}}, \end{array}$$(14c)$$\begin{array}{cc} \; & \;\;\;\;\;\;\;\;\;\;\;\;\;\;\;\sum_{k=1}^{K}{∥w_{k}∥}^{2}+{∥w_{e}∥}^{2}\leq P_{\max}, \end{array}$$(14d)$$\begin{array}{cc} \; & \;\;\;\;\;\;\;\;\;\;\;\;\;\;\;\left|\theta_{m}\right|=1,m\in M, \end{array}$$(14e)$$\begin{array}{cc} \; & \;\;\;\;\;\;\;\;\;\;\;\;\;\;\;s_{i}\in A_{i},i\in \left\{k,e\right\},k\in K, \end{array}$$
where $\gamma_{k,\mathrm{th}}$ and $\gamma_{e,\mathrm{th}}$ are the target threshold of SINR and $P_{\max}$ is the total transmit power budget at the BS.

## 3. Joint Beamforming and MA Positions Optimization

The optimization problem involves multiple coupled variables, including the transmit beamforming vectors for both IR users and the IoT node, the RIS reflection matrix, and the spatial locations of MAs. These variables are intricately coupled through non-convex SINR expressions and EH functions, further complicated by the unit-modulus constraints on RIS elements and the non-convex feasible regions for antenna positions. As a result, the formulated problem is highly non-convex and lacks a closed-form solution. To efficiently address this issue, we employ an AO strategy that decomposes the original formulation into three more tractable subproblems. Each subproblem is addressed while keeping the other variables fixed, the procedure continues until a convergence condition or a predefined iteration cap is satisfied.

### 3.1. Active Beamforming Optimization

In this subsection, we focus on optimizing the active beamforming vectors at the BS, assuming that the RIS phase shifts and the positions of the MA are fixed. Under this condition, the EH expression and SINR constraints become functions of the BS beamformers only. To eliminate the quadratic terms in the SINR expressions and facilitate convex reformulation, we define the rank-one Hermitian matrices $W_{i}=w_{i}w_{i}^{H}$ and $H_{i}=h_{i}h_{i}^{H}$, where the equivalent cascaded channel is given by(15)$$h_{i}=h_{s,i}^{H}\Phi H,i=\left\{k,e\right\},k\in K.$$

Using these definitions, the original problem P1 can be equivalently transformed into a simplified form denoted as(16)$$\begin{array}{cc} \; & P2.1:\max_{{\left\{W_{k}\right\}}_{k=1}^{K},W_{e}}\eta \left(1-\beta \right)\left(\sum_{k=1}^{K}\mathrm{tr}\left(H_{k}W_{k}\right)+\mathrm{tr}\left(H_{e}W_{e}\right)\right) \end{array}$$(16a)$$\begin{array}{cc} \; & \;\;\;\;\;\;\;\;\;s.t.\;\frac{\mathrm{tr}\left(H_{k}W_{k}\right)}{\sum_{i\neq k}^{K}\mathrm{tr}\left(H_{k}W_{i}\right)+\mathrm{tr}\left(H_{k}W_{e}\right)+\sigma_{k}^{2}}\geq \gamma_{k,\mathrm{th}},k\in K, \end{array}$$(16b)$$\begin{array}{cc} \; & \;\;\;\;\;\;\;\;\;\;\;\;\;\;\;\frac{\mathrm{tr}\left(H_{e}W_{e}\right)}{\sum_{k=1}^{K}\mathrm{tr}\left(H_{e}W_{k}\right)+\sigma_{e,d}^{2}}\geq \gamma_{e,\mathrm{th}}, \end{array}$$(16c)$$\begin{array}{cc} \; & \;\;\;\;\;\;\;\;\;\;\;\;\;\;\;\sum_{k=1}^{K}\mathrm{tr}\left(W_{k}\right)+\mathrm{tr}\left(W_{e}\right)\leq P_{\max}, \end{array}$$(16d)$$\begin{array}{cc} \; & \;\;\;\;\;\;\;\;\;\;\;\;\;\;\;W_{k}⪰0,W_{e}⪰0,k\in K, \end{array}$$(16e)$$\begin{array}{cc} \; & \;\;\;\;\;\;\;\;\;\;\;\;\;\;\;\mathrm{rank}\left(W_{k}\right)=1,\mathrm{rank}\left(W_{e}\right)=1,k\in K. \end{array}$$

The rank-one condition in (Equation 21e) makes problem P2.1 inherently non-convex, preventing a direct solution. To address this challenge, we adopt the semidefinite relaxation (SDR) technique to remove the rank restriction and cast the problem into a conventional semidefinite programming (SDP) form. The resulting relaxed problem can be efficiently solved using convex optimization toolboxes such as CVX. The tightness of this relaxation is examined through the following theoretical analysis.

**Theorem** **1.**
*Suppose the SDR of problem P2.1, where the rank-one constraint in (Equation 21e) is omitted, is feasible. Then, the optimal solutions $W_{k}$ and $W_{e}$ to the relaxed problem satisfy $\mathrm{rank}\left(W_{k}\right)\leq 1$ and $\mathrm{rank}\left(W_{e}\right)\leq 1$, for every k∈K.*


**Proof.** The proof begins by constructing the Lagrangian of problem P2.1, as follows:(17)$$\begin{array}{c} L\left(\left\{W_{k}\right\},W_{e},\left\{\lambda_{k}\right\},\mu ,\nu ,\left\{S_{k}\right\},S_{e}\right)=\eta \left(1-\beta \right)\left(\sum_{k=1}^{K}\mathrm{tr}\left(H_{k}W_{k}\right)+\mathrm{tr}\left(H_{e}W_{e}\right)\right)\\ \;+\sum_{k=1}^{K}\lambda_{k}\left(\mathrm{tr}\left(H_{k}W_{k}\right)-\gamma_{k,\mathrm{th}}\left[\sum_{i\neq k}\mathrm{tr}\left(H_{k}W_{i}\right)+\mathrm{tr}\left(H_{k}W_{e}\right)+\sigma_{k}^{2}\right]\right)\\ \;+\mu \left(\mathrm{tr}\left(H_{e}W_{e}\right)-\gamma_{e,\mathrm{th}}\left[\sum_{k=1}^{K}\mathrm{tr}\left(H_{e}W_{k}\right)+\sigma_{e,d}^{2}\right]\right)\\ \;-\nu \left(\sum_{k=1}^{K}\mathrm{tr}\left(W_{k}\right)+\mathrm{tr}\left(W_{e}\right)-P_{\max}\right)\\ \;-\sum_{k=1}^{K}S_{k}W_{k}-S_{e}W_{e}. \end{array}$$
where the associated variables satisfy ${\left\{\lambda_{k}\right\}}_{k=1}^{K}\geq 0$, μ≥0, ν≥0, ${\left\{S_{k}\right\}}_{k=1}^{K}⪰0$, and $\left\{S_{e}\right\}⪰0$. Subsequently, we compute the partial derivative of (Equation 27) with respect to $W_{n}$, k∈K, and obtain the corresponding KKT optimality condition as follows:(18)$$\begin{array}{c} \left(\eta \left(1-\beta \right)+\lambda_{k}\right)H_{k}-\sum_{j\neq k}^{K}\lambda_{j}\gamma_{j,\mathrm{th}}H_{j}-\mu \gamma_{e,\mathrm{th}}H_{e}-\nu I-S_{k}=0, \end{array}$$(19)$$\begin{array}{c} S_{k}W_{k}=0, \end{array}$$(20)$$\begin{array}{c} W_{k}⪰0. \end{array}$$It is evident that in Equation (Equation 28), the matrix $A=\sum_{j\neq k}^{K}\lambda_{j}\gamma_{j,\mathrm{th}}H_{j}+\mu \gamma_{e,\mathrm{th}}H_{e}+\nu I$ is positive, definite, and of full-rank. The proof is straightforward: since $\lambda_{j}$, μ, and ν are all nonnegative, and both $H_{j}$ and $H_{e}$ are rank-one matrices, the inclusion of the full-rank term νI ensures that A is strictly positive definite and full-rank. Then, by right-multiplying (Equation 28) with $W_{k}$ and substituting Equation (Equation 29), we obtain(21)$$\left(\eta \left(1-\beta \right)+\lambda_{k}\right)H_{k}W_{k}=AW_{k}.$$Based on this relation, it follows that(22)$$\mathrm{rank}\left(AW_{k}\right)=\mathrm{rank}\left(W_{k}\right)=\mathrm{rank}\left(\left(\eta \left(1-\beta \right)+\lambda_{k}\right)H_{k}W_{k}\right)\leq 1.$$Therefore, we conclude that $\mathrm{rank}\left(W_{k}\right)\leq 1$. Similarly, it can be shown that $\mathrm{rank}\left(W_{e}\right)\leq 1$. □

### 3.2. Passive Beamforming Optimization

After optimizing the active beamforming vectors at the BS and fixing the positions of the MAs, we proceed to optimize the RIS phase shifts Φ. To handle the non-convex and nonlinear dependence of the objective and constraints on the RIS phase shifts, we introduce an equivalent reformulation by expressing the effective channels in terms of the RIS parameters. Specifically, we define $H_{i,j}=h_{i,j}h_{i,j}^{H}$ with $i,j=\left\{s,e\right\}$, $\Phi =diag\left(q\right)$, and $Q=qq^{H}$. The details are provided as(23)$$h_{i,j}^{H}=h_{s,i}^{H}\mathrm{diag}\left(Hw_{j}\right),i,j=\left\{k,e\right\},k\in K,$$
and(24)$$q=\left[\theta_{1},\theta_{2},\ldots ,\theta_{M}\right].$$

At this point, P1 can be restructured as(25)$$\begin{array}{cc} \; & P3:\max_{Q}\eta \left(1-\beta \right)\left(\sum_{k=1}^{K}\mathrm{tr}\left(H_{e,k}Q\right)+\mathrm{tr}\left(H_{e,e}Q\right)\right) \end{array}$$(25a)$$\begin{array}{cc} \; & \;\;\;\;\;\;\;\;\;s.t.\;\frac{\mathrm{tr}\left(H_{k,k}Q\right)}{\sum_{i\neq k}^{K}\mathrm{tr}\left(H_{k,i}Q\right)+\mathrm{tr}\left(H_{k,e}Q\right)+\sigma_{k}^{2}}\geq \gamma_{k,\mathrm{th}},k\in K, \end{array}$$(25b)$$\begin{array}{cc} \; & \;\;\;\;\;\;\;\;\;\;\;\;\;\;\;\frac{\mathrm{tr}\left(H_{e,e}Q\right)}{\sum_{k=1}^{K}\mathrm{tr}\left(H_{e,k}Q\right)+\sigma_{e,d}^{2}}\geq \gamma_{e,\mathrm{th}}, \end{array}$$(25c)$$\begin{array}{cc} \; & \;\;\;\;\;\;\;\;\;\;\;\;\;\;\;Q_{m,m}=1,m\in M, \end{array}$$(25d)$$\begin{array}{cc} \; & \;\;\;\;\;\;\;\;\;\;\;\;\;\;\;Q⪰0, \end{array}$$(25e)$$\begin{array}{cc} \; & \;\;\;\;\;\;\;\;\;\;\;\;\;\;\;\mathrm{rank}\left(Q\right)=1, \end{array}$$

Because constraint (Equation 40) enforces a rank-one structure, problem P3 is inherently non-convex. To bypass this difficulty, we apply a semidefinite relaxation that discards the rank requirement and converts the formulation into a tractable SDP, solvable using standard convex solvers such as CVX. When the optimized matrix Q is not rank-one, a valid RIS phase vector is subsequently generated through Gaussian randomization.

### 3.3. MA Positions Optimization

With the active and passive beamforming vectors fixed, we now focus on optimizing the position parameters of the MAs, which directly influence the channel vectors between the MAs and both the RIS and the receivers. Then, the subproblem is reformulated as(26)$$\begin{array}{cc} \; & P4:\max_{{\left\{s_{k}\right\}}_{k=1}^{K},s_{e}}E_{h} \end{array}$$(26a)$$\begin{array}{cc} \; & \;\;\;\;\;\;\;\;\;s.t.\;\gamma_{k}\geq \gamma_{k,\mathrm{th}},k\in K, \end{array}$$(26b)$$\begin{array}{cc} \; & \;\;\;\;\;\;\;\;\;\;\;\;\;\;\;\gamma_{e}\geq \gamma_{e,\mathrm{th}}, \end{array}$$(26c)$$\begin{array}{cc} \; & \;\;\;\;\;\;\;\;\;\;\;\;\;\;\;s_{i}\in A_{i},i\in \left\{k,e\right\},k\in K, \end{array}$$

To facilitate a more tractable formulation, we define an intermediate matrix $\Omega_{si}$, which isolates the beamforming-related terms independent of the antenna positions. Specifically, for all nodes, we define(27)$$\Omega_{i,j}=\Sigma_{si}^{H}F_{si}\Phi Hw_{j}w_{j}^{H}H^{H}\Phi^{H}F_{si}^{H}\Sigma_{si},i,j\in \left\{k,e\right\},k\in K.$$

Substituting this expression into the EH function, the original objective can be reformulated into a more position-explicit form:(28)$$E_{h}=\eta \left(1-\beta \right)g_{se}^{H}\left(s_{e}\right)\left(\sum_{k=1}^{K}\Omega_{e,k}+\Omega_{e,e}\right)g_{se}\left(s_{e}\right).$$

To further highlight the role of the MA positions in the objective function, we explicitly express the position-dependent terms using the following forms. Thus, the quadratic terms can be rewritten as(29)$$E_{h}\left(s_{e}\right)=\eta \left(1-\beta \right)\sum_{i=1}^{L_{e}^{r}}\sum_{j=1}^{L_{e}^{r}}\left|{\left[\Omega_{e}\right]}_{i,j}\right|\cos\left(\frac{2\pi }{\lambda }\rho_{e,j}^{r}\left(s_{e}\right)-\frac{2\pi }{\lambda }\rho_{e,i}^{r}\left(s_{e}\right)+∠{\left[\Omega_{e}\right]}_{i,j}\right),$$
where $\Omega_{e}=\sum_{k=1}^{K}\Omega_{e,k}+\Omega_{e,e}$. However, the expressions $E_{h}\left(s_{e}\right)$ is non-convex with respect to the MA position variables due to the embedded trigonometric terms. To facilitate tractable optimization, we adopt the second-order Taylor expansion to approximate each term with a concave lower bound, enabling the use of SCA techniques. Specifically, for a given point $s_{e}$, the function $E_{h}\left(s_{e}\right)$ is approximated:(30)$$E_{h}\left(s_{e}\right)\geq E_{h}\left({\bar{s}}_{e}\right)+\nabla E_{h}{\left({\bar{s}}_{e}\right)}^{T}\left(s_{e}-{\bar{s}}_{e}\right)-\frac{\delta_{e}}{2}{\left(s_{e}-{\bar{s}}_{e}\right)}^{T}\left(s_{e}-{\bar{s}}_{e}\right)\overset{\Delta }{\;=}{\bar{E}}_{h}\left(s_{e}\right),$$
where $\delta_{e}$ is upper bound on the curvature of $E_{h}\left(s_{e}\right)$ with $\delta_{e}I_{2}⪰\nabla^{2}E_{h}\left(s_{e}\right)$. The gradient and Hessian matrixes for $E_{h}\left(s_{e}\right)$ are provided in Appendix A. Based on these, we can obtain(31)$$\begin{array}{cc} {∥\nabla^{2}E_{h}\left(s_{e}\right)∥}_{2}^{2} & \leq {∥\nabla^{2}E_{h}\left(s_{e}\right)∥}_{F}^{2}={\left(\frac{\partial^{2}E_{h}\left(s_{e}\right)}{\partial x_{e}^{2}}\right)}^{2}+{\left(\frac{\partial^{2}E_{h}\left(s_{e}\right)}{\partial y_{e}^{2}}\right)}^{2}+2{\left(\frac{\partial^{2}E_{h}\left(s_{e}\right)}{\partial x_{e}\partial y_{e}}\right)}^{2}\\ & \leq 4{\left(\frac{16\pi^{2}}{\lambda^{2}}\sum_{i=1}^{L_{e}^{r}}\sum_{j=1}^{L_{e}^{r}}\left|{\left[\Omega_{e}\right]}_{i,j}\right|\right)}^{2}. \end{array}$$

Therefore, we can obtain an upper bound $\delta_{e}$ for the curvature(32)$$\delta_{e}=\frac{32\pi^{2}}{\lambda^{2}}\sum_{i=1}^{L_{e}^{r}}\sum_{j=1}^{L_{e}^{r}}\left|{\left[\Omega_{e}\right]}_{i,j}\right|.$$

At this point, after replacing ${\bar{E}}_{h}\left(s_{e}\right)$ with its concave lower bound $E_{h}\left(s_{e}\right)$, the resulting objective function becomes convex with respect to the MA position variable $s_{e}$. However, constraints (Equation 42) and (Equation 43) still contain nonlinear coupling terms, requiring similar relaxation strategies for efficient solution. Similarly, we first rewrite the SINR expressions in an equivalent quadratic form.(33)$$\gamma_{k}\left(s_{k}\right)=\frac{1}{\sigma_{k}^{2}}g_{sk}^{H}\left(s_{k}\right)\left(\Omega_{k,k}-\gamma_{k,\mathrm{th}}\left(\sum_{i\neq k}^{K}\Omega_{k,i}+\Omega_{k,e}\right)\right)g_{sk}\left(s_{k}\right)\geq \gamma_{k,\mathrm{th}},k\in K,$$
and(34)$$\gamma_{e}\left(s_{e}\right)=\frac{1}{\sigma_{e,d}^{2}}g_{se}^{H}\left(s_{e}\right)\left(\Omega_{e,e}-\gamma_{e,\mathrm{th}}\sum_{k=1}^{K}\Omega_{e,k}\right)g_{se}\left(s_{e}\right)\geq \gamma_{e,\mathrm{th}}.$$

To further make the dependence on MA positions explicit, we express the SINRs as(35)$$\gamma_{k}\left(s_{k}\right)=\sum_{i=1}^{L_{e}^{r}}\sum_{j=1}^{L_{e}^{r}}\left|{\left[{\bar{\Omega }}_{k}\right]}_{i,j}\right|\cos\left(\frac{2\pi }{\lambda }\rho_{k,j}^{r}\left(s_{k}\right)-\frac{2\pi }{\lambda }\rho_{k,i}^{r}\left(s_{k}\right)+∠{\left[{\bar{\Omega }}_{k}\right]}_{i,j}\right),k\in K,$$
and(36)$$\gamma_{e}\left(s_{e}\right)=\sum_{i=1}^{L_{e}^{r}}\sum_{j=1}^{L_{e}^{r}}\left|{\left[{\bar{\Omega }}_{e}\right]}_{i,j}\right|\cos\left(\frac{2\pi }{\lambda }\rho_{e,j}^{r}\left(s_{e}\right)-\frac{2\pi }{\lambda }\rho_{e,i}^{r}\left(s_{e}\right)+∠{\left[{\bar{\Omega }}_{e}\right]}_{i,j}\right).$$
where ${\bar{\Omega }}_{k}=\frac{1}{\sigma_{k}^{2}}\left(\Omega_{k,k}-\gamma_{k,\mathrm{th}}\left(\sum_{i\neq k}^{K}\Omega_{k,i}+\Omega_{k,e}\right)\right)$ and ${\bar{\Omega }}_{e}=\frac{1}{\sigma_{e,d}^{2}}\left(\Omega_{e,e}-\gamma_{e,\mathrm{th}}\sum_{k=1}^{K}\Omega_{e,k}\right)$. Since these expressions are still non-convex, we apply the second-order Taylor expansion to obtain their concave lower bounds as follows:(37)$$\gamma_{k}\left(s_{k}\right)\geq \gamma_{k}\left({\bar{s}}_{k}\right)+\nabla \gamma_{k}{\left({\bar{s}}_{k}\right)}^{T}\left(s_{k}-{\bar{s}}_{k}\right)-\frac{{\bar{\delta }}_{k}}{2}{\left(s_{k}-{\bar{s}}_{k}\right)}^{T}\left(s_{k}-{\bar{s}}_{k}\right)\geq \gamma_{k,\mathrm{th}},k\in K,$$
and(38)$$\gamma_{e}\left(s_{e}\right)\geq \gamma_{e}\left({\bar{s}}_{e}\right)+\nabla \gamma_{e}{\left({\bar{s}}_{e}\right)}^{T}\left(s_{e}-{\bar{s}}_{e}\right)-\frac{{\bar{\delta }}_{e}}{2}{\left(s_{e}-{\bar{s}}_{e}\right)}^{T}\left(s_{e}-{\bar{s}}_{e}\right)\geq \gamma_{e,\mathrm{th}}.$$

The gradients and curvature upper bounds ${\bar{\delta }}_{k}$ and ${\bar{\delta }}_{e}$, which are required for constructing the concave lower bounds of $\gamma_{k}\left(s_{k}\right)$ and $\gamma_{e}\left(s_{e}\right)$, can be easily computed in a similar manner to those given in Equations (Equation 57)–(Equation 50). For brevity, we omit their explicit expressions here. At this point, the original non-convex problem P4 has been transformed into a tractable convex approximation through the use of the SCA. The resulting optimization problem can be efficiently solved using standard convex solvers such as the CVX.

### 3.4. Convergence and Complexity Analysis

To conclude, Algorithm 1 provides the detailed steps of the SCA-based AO scheme. We further investigate its convergence properties along with the associated computational burden.
**Algorithm 1** Alternating optimization scheme for beamforming and MA position design.1:**Initialization**: Set maximum iteration $T_{\max}$, tolerance ε, initial index iter=1, initial values ${\left\{w_{k}^{\left(0\right)}\right\}}_{k=1}^{K},w_{e}^{\left(0\right)},\Phi^{\left(0\right)},{\left\{s_{k}^{\left(0\right)}\right\}}_{k=1}^{K},s_{e}^{\left(0\right)}$.2:**repeat**3: Given Φ, ${\left\{s_{k}\right\}}_{k=1}^{K}$, and $s_{e}$, obtain ${\left\{w_{k}\right\}}_{k=1}^{K}$ and $w_{e}$ by P2;4: Given ${\left\{w_{k}^{\left(iter\right)}\right\}}_{k=1}^{K}$, $w_{e}^{\left(iter\right)}$, and $s_{e}$, obtain $\Phi^{\left(iter\right)}$ by *P*3;5: Update ${\left\{w_{k}^{\left(iter\right)}\right\}}_{k=1}^{K}$, $w_{e}^{\left(iter\right)}$, and $\Phi^{\left(iter\right)}$, obtain ${\left\{s_{k}^{\left(iter\right)}\right\}}_{k=1}^{K}$ and $s_{e}^{\left(iter\right)}$ by *P*4;6: Update iter=iter+1;7:**until** the increase in the objective value is below ε or the number of iterations exceeds $T_{\max}$.8:**return** ${\left\{w_{k}\right\}}_{k=1}^{K}$, $w_{e}$, Φ, ${\left\{s_{k}\right\}}_{k=1}^{K}$, and $s_{e}$.

#### 3.4.1. Convergence Analysis

In each iteration, one set of variables is optimized while fixing the others, and the non-convex subproblems are replaced by their tractable convex approximations. Because the objective value does not decrease across iterations and the transmit power remains bounded, the iterative procedure is ensured to converge.

#### 3.4.2. Complexity Analysis

The computational burden of the entire procedure is mainly determined by the complexity of handling subproblems *P*2, *P*3, and *P*4. According to [31], the per-iteration complexity of solving *P*2 can be expressed as $O(N^{4.5}K^{3.5})$. For subproblem *P*3, which updates the RIS phase-shift vector, the computational cost scales on the order of $O(M^{4.5})$. The computational complexity of solving *P*4 is given by $O(K^{3.5})$. Thus, the algorithm remains tractable for practical-scale IoT applications with moderate *N*, *M*, and *K*.

## 4. Results

To assess the performance of the MA-assisted RIS–SWIPT system, we configure the simulation environment according to the parameter settings in [23,30]. In our simulations, all variables are initialized randomly without requiring problem-specific tuning or fixed starting points. The RIS is placed 30 m away from the BS, while the legitimate user and IoT energy-harvesting node are distributed at the RIS at a distance of 10 m but located at different angular positions. For fairness and consistency, all involved wireless links (i.e., BS-to-RIS, RIS-to-user, BS-to-IoT, etc.) are assumed to experience the same number of propagation paths, denoted by *L*, such that $L_{i}^{r}=L_{i}^{t}=L_{b}^{r}=L_{b}^{t}=L$, where $i\in \left\{k,e\right\},k\in K$. The path response matrix of each channel follows the diagonal matrix model. Specifically, the first diagonal entry follows a complex Gaussian distribution ${\left[\Sigma_{p}\right]}_{11}∼\mathrm{CN}\left(0,g_{0}d_{p}^{-2}\kappa_{p}/\left(\kappa_{p}+1\right)\right)$, while the remaining diagonal entries are distributed as ${\left[\Sigma_{p}\right]}_{jj}∼\mathrm{CN}\left(0,g_{0}d_{p}^{-2}/\left(\left(L-1\right)\left(\kappa_{p}+1\right)\right)\right),j=2,3,\ldots ,L$, where $p\in \left\{sr,bi\right\}$, $i\in \left\{k,e\right\}$, k∈K. Moreover, $g_{0}=-30\;\mathrm{dB}$ represents the average large-scale channel power at a reference distance of 1 m. The Rician factors are set to $\kappa_{sr}=2$ for the BS-to-RIS link, and $\kappa_{bi}=2.7$ for both the user and IoT nodes. All AWGN introduced at each stage is uniformly set to −90 dBm. Moreover, the side length of movable region $A_{i}$ for all MAs is set to be consistent, i.e., $D_{i}=D$, $i\in \left\{k,e\right\},k\in K$. The energy conversion efficiency is not considered in the simulations, and hence the efficiency factor is set to η=1 to isolate the effects of our proposed optimization strategy. All simulations were performed on MATLAB and the CVX toolbox.

*Tag/Benchmark:* We denote the proposed MA-enabled RIS-SWIPT scheme as “Proposed MA”. To evaluate the impact of MA on system performance, we adopt the FPA deployment as a baseline, which serves as a widely used baseline in RIS-assisted SWIPT literature, referred to as "FPA". In addition, a scenario with randomly configured RIS phase shifts serves as a benchmark for evaluating the effectiveness of phase-shift optimization.

The convergence behavior of the SCA-based scheme for multiple scenarios is depicted in Figure 2. The harvested power $E_{h}$ increases rapidly within the first few iterations and gradually converges within 15 iterations, verifying both the efficiency and rapid convergence of the algorithm. Moreover, even with increased *M* and *N*, the proposed algorithm still exhibits fast convergence, demonstrating its robustness and scalability.

In Figure 3, the harvested energy is plotted against the number of RIS elements *M*. An increase in *M* consistently improves $E_{h}$ across all schemes because more elements yield stronger passive beamforming capability. The proposed MA scheme consistently outperforms the FPA baseline, and both significantly exceed the random-phase benchmark. This confirms the advantage of MA-enabled spatial diversity and optimized RIS phase shifts.

The relationship between harvested power and the transmit antenna count is plotted in Figure 4. It can be observed that increasing *N* significantly improves system performance across all schemes. This is because more antennas provide higher DoF for active beamforming, enabling more effective exploitation of the multi-path components from the BS to RIS. The MA-based design delivers the highest harvested power, exceeding both the FPA and random-phase benchmarks, with its advantage becoming more evident as *N* grows.

Figure 5 depicts how the harvested power varies with the transmit power. As expected, $E_{h}$ increases with higher $P_{\max}$ for all schemes. The proposed MA scheme consistently outperforms the benchmark methods, and the performance gap becomes more significant at high power levels. This demonstrates the superior energy transfer efficiency of the MA architecture, especially in high-power regimes. Notably, the FPA with random phase remains ineffective across the entire power range, highlighting the importance of joint optimization in RIS-based SWIPT systems.

Figure 6 illustrates the harvested power vs. the normalized moving range. As D/λ increases, the harvested power improves under the proposed MA scheme, owing to enhanced spatial DoF that enables better channel adaptation. However, the performance gain becomes marginal when D/λ>0.6, indicating that substantial improvements can already be achieved with moderate movement ranges. This demonstrates that the proposed method does not rely on large mobility, making it more practical for implementation.

Figure 7 shows the harvested power as a function of the power-splitting ratio β, which characterizes the portion of received power allocated to information decoding at IoT nodes. As expected, increasing β reduces the portion of power available for EH, thereby degrading $E_{h}$. The proposed MA scheme consistently outperforms the benchmarks under all values of β, demonstrating its robustness under different communication–energy tradeoff settings. Notably, even when more power is allocated to information decoding (β≥0.5), the proposed method still achieves significant performance gain over FPA.

## 5. Conclusions

This paper explored an RIS-assisted SWIPT system empowered by movable antennas. A unified optimization model has been established to maximize the energy harvested at IoT devices by adjusting the BS beamforming, RIS phase shifts, and MA positions under communication constraints. To handle the intrinsic non-convexity of the formulation, an AO strategy leveraging SCA and second-order Taylor approximations was developed. Numerical assessments indicated that the proposed MA-enabled strategy produces substantial enhancements in EH performance compared to conventional FPA-based benchmarks. Furthermore, we have shown that the performance gain remains substantial even with limited MA movement, confirming the practicality and effectiveness of the proposed approach for future green IoT deployments.

This work focuses on the theoretical performance of MA–RIS-assisted SWIPT systems. In the future, we will investigate the practical challenges associated with MA implementations, including actuation latency, mechanical constraints, and energy overhead to bridge theory and implementation. It is of great interest to further investigate the hardware constraints associated with the integration of MA and RIS, such as positioning accuracy, control signaling latency, and limitations in phase-shift resolution. Moreover, we will consider integrating hybrid beamforming strategies into the proposed MA-RIS framework and conducting studies with other emerging metasurface-assisted architectures.

## Figures and Tables

**Figure 1 sensors-25-07402-f001:**
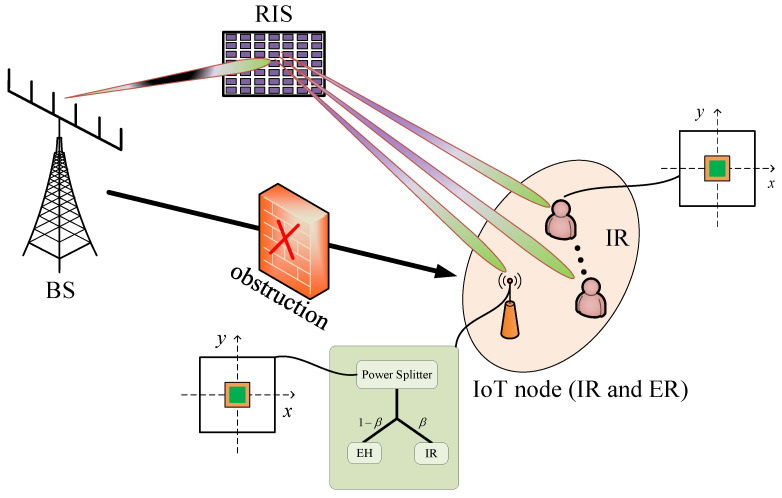
System model.

**Figure 2 sensors-25-07402-f002:**
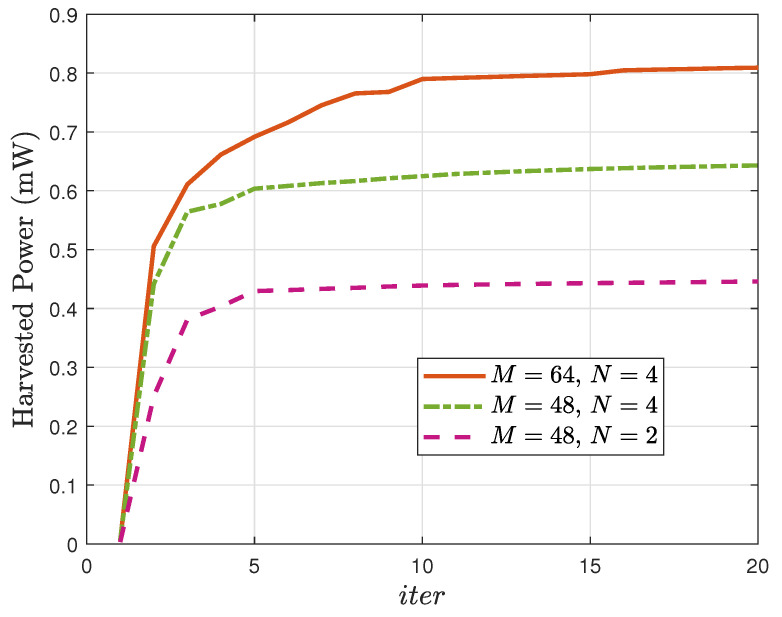
Harvested power $E_{h}$ vs. the number of iterations iter.

**Figure 3 sensors-25-07402-f003:**
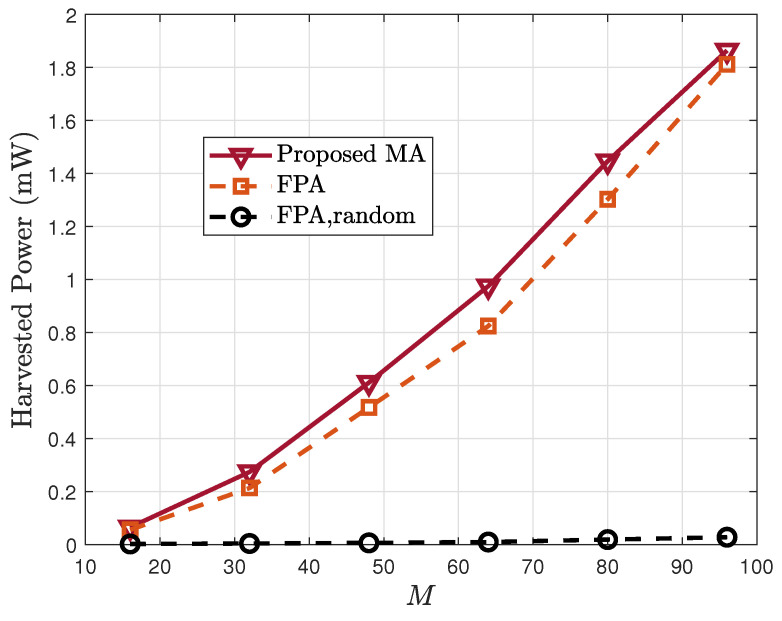
Harvested power $E_{h}$ vs. the number of RIS elements *M*.

**Figure 4 sensors-25-07402-f004:**
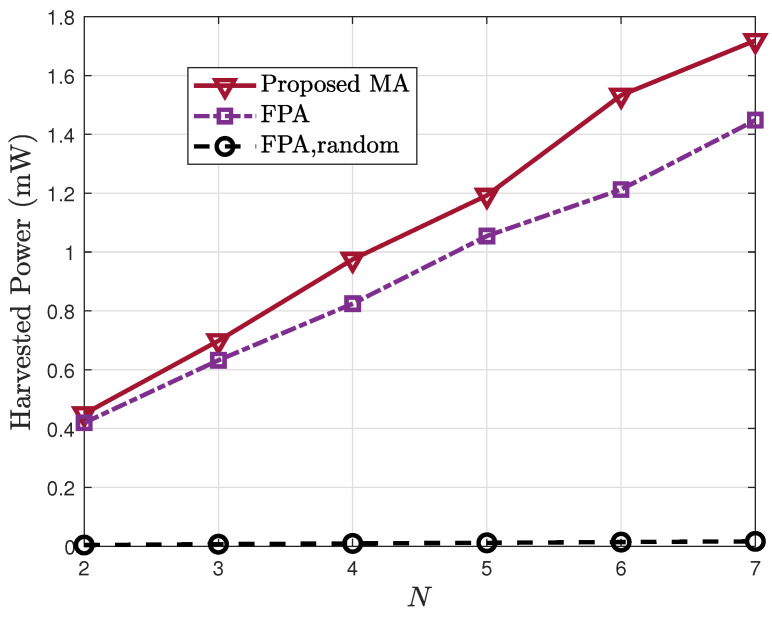
Harvested power $E_{h}$ vs. the number of transmit antennas *N*.

**Figure 5 sensors-25-07402-f005:**
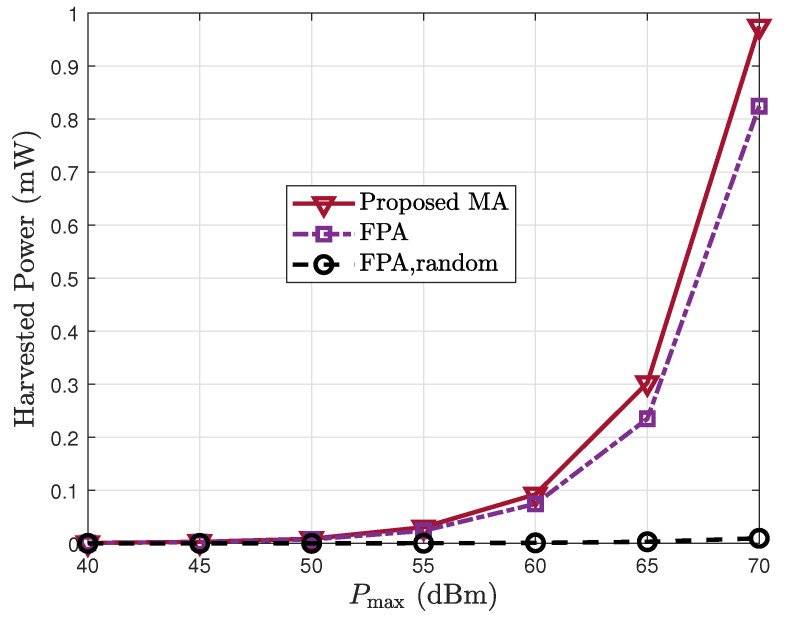
Harvested power $E_{h}$ vs. the transmit power $P_{\max}$.

**Figure 6 sensors-25-07402-f006:**
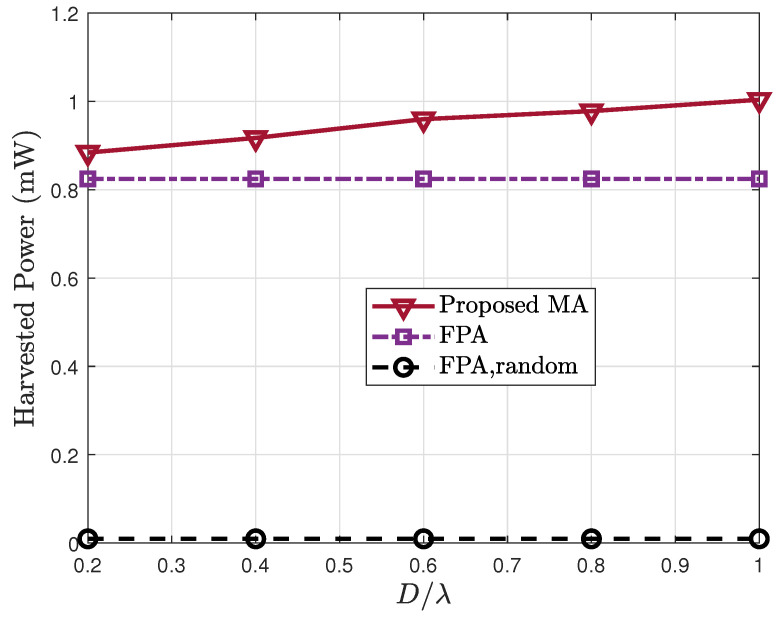
Harvested power $E_{h}$ vs. the normalized moving range D/λ.

**Figure 7 sensors-25-07402-f007:**
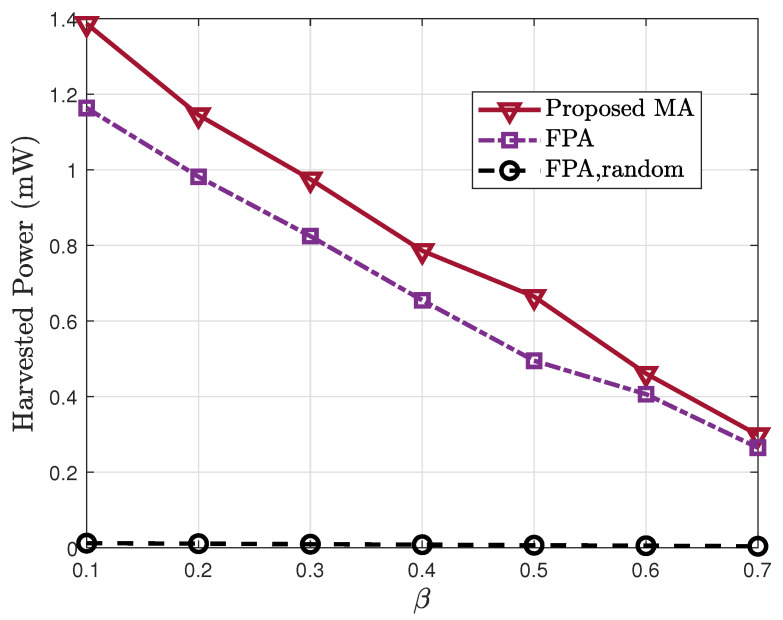
Harvested power $E_{h}$ vs. the power-splitting ratio β.

**Table 1 sensors-25-07402-t001:** List of frequently used acronyms in this paper.

Acronym	Full Term
AO	Alternating Optimization
BS	Base Station
EH	Energy Harvesting
IoT	Internet of Things
MA	Movable Antenna
RIS	Reconfigurable Intelligent Surface
SCA	Successive Convex Approximation
SDR	Semidefinite Relaxation
SINR	Signal-to-Interference-plus-Noise Ratio
SWIPT	Simultaneous Wireless Information and Power Transfer

## Data Availability

The raw data supporting the conclusions of this paper will be made available by the authors upon reasonable request.

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
