# Peer review of "Movable Antenna-Enabled RIS-Assisted Simultaneous Wireless Information and Power Transfer Systems"

_sensors, 2025, doi:10.3390/s25237402_

Round 1
Reviewer 1 Report
Comments and Suggestions for Authors
- In Section 2, it is assumed that the system has a single IoT node. However, “IoT energy-harvesting nodes”appear in the simulation experiments. Please explain how to expand the system model from “single node” to “nodes”.
- Please explain how the values of the simulation parameters in Section 4 are chosen.
- In Section 4, some simulation results are given for different parameter values. Are there any results in literature that can be referred for comparison except the conventional FPA-based scheme?
- Please correct the mistakes in the main text, for example, duplicated “both”in line 226.
Author Response
|
1. Summary |
|
|
|
We sincerely thank you for the careful reading and thoughtful comments. The reviewer's suggestions have greatly helped us improve the clarity and completeness of our manuscript. In particular, the reviewer raised important points regarding the consistency of the system model description, the rationale behind simulation parameter selection, comparative baselines, and minor textual errors. We have addressed each of your comments in detail below, with corresponding revisions marked in the revised manuscript.
|
||
|
2. Point-by-point response to Comments and Suggestions for Authors
|
||
|
Comments 1: In Section 2, it is assumed that the system has a single IoT node. However, “IoT energy-harvesting nodes”appear in the simulation experiments. Please explain how to expand the system model from “single node” to “nodes”.
|
||
|
Response 1: Thank you for your careful reading and valuable suggestion. We sincerely apologize for the confusion caused by the inconsistent wording. As correctly pointed out, the system model in Section 2 is built for a single IoT energy-harvesting node, and this assumption is consistently maintained throughout the optimization and algorithm design. The phrase “IoT energy-harvesting nodes” in the simulation section was a typographical error, and we have corrected this to the singular form in the revised manuscript to ensure consistency. Furthermore, we agree with the reviewer that extending the proposed framework to support multiple IoT nodes is both practical and meaningful. We would like to clarify that: 1) The proposed formulation and algorithm are inherently compatible with multiple IoT nodes. 2) Specifically, the objective function can be modified to include the sum of harvested power across multiple nodes, while the convex approximation and alternating optimization structure remain applicable. 3) Such extension will not change the convexity properties of each subproblem nor the convergence of the proposed algorithm. Therefore, we have added a brief explanation regarding this possible extension in Section 2 of the revised manuscript to guide future implementation, as follows: (Section 2) Although the present work examines a single IoT node, the adopted modeling methodology is flexible and can be extended to configurations with multiple IoT nodes. Furthermore, the algorithm can be adapted to multi IoT nodes with only minor adjustments.
Comments 2: Please explain how the values of the simulation parameters in Section 4 are chosen. Response 2: Thank you for your valuable comment. The selection of simulation parameters in Section 4 is primarily based on widely accepted settings in related literature, such as [17] and [24]. These references provide typical values for path loss, transmission power, RIS configurations, and energy harvesting models in similar SWIPT system studies. To improve clarity, we have now added a corresponding explanation in in Section 4 of the revised manuscript, as follows: (Section 4) To evaluate the performance of the proposed MA-enabled RIS-SWIPT system, simulation parameters are adopted with the following setup [17] and [24].
Comments 3: In Section 4, some simulation results are given for different parameter values. Are there any results in literature that can be referred for comparison except the conventional FPA-based scheme?
Response 3: We sincerely appreciate your constructive comment. We would like to clarify that the manuscript already includes two representative baseline schemes in the simulation section: 1) FPA-based scheme: This represents the conventional fixed-position antenna scenario, serving as a widely used baseline in RIS-assisted SWIPT literature. 2) Random RIS configuration: This serves as a benchmark for evaluating the effectiveness of phase shift optimization by comparing against non-optimized RIS elements. These two baselines were selected to highlight the contributions of our proposed MA-enabled joint optimization framework. As the integration of movable antennas and RIS in SWIPT systems remains relatively underexplored, few existing works provide directly comparable results. To clarify our chosen benchmarking approach, the revised manuscript includes supplementary explanations clarifying the rationale for their selection. (Section 4) To evaluate the impact of MA on system performance, we adopt the FPA deployment as a baseline, which serves as a widely used baseline in RIS-assisted SWIPT literature, referred to as "FPA". In addition, a scenario with randomly configured RIS phase shifts serves as a benchmark for evaluating the effectiveness of phase shift optimization.
Comments 4: Please correct the mistakes in the main text, for example, duplicated “both”in line 226.
Response 4: Thank you very much for pointing out the typo. We have carefully reviewed the manuscript and corrected the duplicated "both" in line 226. In addition, we have re-checked the entire manuscript and made minor textual corrections to improve clarity and grammar consistency. |
||
Thank you again for your generosity and the valuable time you invested in improving our submission. We hope that the modifications and clarifications provided address your concerns satisfactorily.
Best regards,
< All authors>

Reviewer 2 Report
Comments and Suggestions for Authors
This is an innovative and timely contribution that introduces a novel approach for enhancing RIS-assisted SWIPT systems through movable antennas. The joint optimization framework and the proposed AO-SCA algorithm are well-developed and validated. However, to strengthen the practical applicability of the work, we suggest discussing the computational complexity, energy conversion efficiency, and hardware limitations in more depth. Overall, the manuscript is clearly written, well-structured, and offers significant value to the IoT and wireless communications community.
Author Response
|
1. Summary |
|
|
We sincerely appreciate your positive and encouraging comments. We are glad to know that you found our work innovative, timely, and valuable to the IoT and wireless communications community. We also thank you for the constructive suggestions to improve the practical relevance of our work. In the revised version, we have carefully addressed your comments regarding computational complexity, energy conversion efficiency, and hardware limitations.
|
2. Point-by-point response to Comments and Suggestions for Authors
|
|
Comments: This is an innovative and timely contribution that introduces a novel approach for enhancing RIS-assisted SWIPT systems through movable antennas. The joint optimization framework and the proposed AO-SCA algorithm are well-developed and validated. However, to strengthen the practical applicability of the work, we suggest discussing the computational complexity, energy conversion efficiency, and hardware limitations in more depth. Overall, the manuscript is clearly written, well-structured, and offers significant value to the IoT and wireless communications community.
Response: Thank you for recognizing the novelty and value of our work. Moreover, we will address each point of your feedback as follows. 1) Computational complexity We have thoroughly analyzed the computational complexity of the proposed AO-based framework in the revised manuscript, as follows: (Section 3) The overall computational complexity of the proposed algorithm is primarily dominated by solving P2, P3, and P4. According to [25], the per-iteration complexity of solving P2 can be expressed as . The computational complexity of solving P3, which optimizes over the RIS phase shifts, is given by . The computational complexity of solving P4 is given by . Thus, the algorithm remains tractable for practical-scale IoT applications with moderate N, M, and K. 2) Energy conversion efficiency As rightly pointed out, the energy harvesting efficiency depends on multiple factors such as rectifier non-linearity, circuit design, and input power dynamics. In this work, we adopt the linear EH model, which is widely used in existing literature to enable tractable analysis and fair algorithm benchmarking (e.g. [24]). Specifically, we use the linear relationship:
In practice, can be readily scaled into the final performance results. Moreover, using does not change the optimization structure, but linearly scales down the harvested energy, which can be compensated by tuning or RIS phase shifts. We have added clarification in the revised manuscript accordingly. (Section 4) The energy conversion efficiency is not considered in the simulations, and hence the efficiency factor is set to to isolate the effects of our proposed optimization strategy. 3) Hardware limitations Thank you for your insightful suggestion regarding hardware constraints. We fully agree that practical limitations, including actuator accuracy, control signaling delay, and phase resolution, may impact the performance of movable antennas and RIS in real-world implementations. To address this, we have added a brief discussion in the conclusion section (Future Work) to highlight this direction for future research. (Section 5) It is of great interest to further investigate the hardware constraints associated with the integration of movable antennas and RIS, such as positioning accuracy, control signaling latency, and limitations in phase shift resolution. Once again, we sincerely thank you for the insightful suggestions that have helped improve the completeness and applicability of our work. |
We sincerely thank you for the encouraging recognition and constructive suggestions. These valuable comments have greatly helped us refine the manuscript.
Best regards,
< All authors>

Reviewer 3 Report
Comments and Suggestions for Authors
The paper presents a well-structured and technically sound framework that combines RIS and movable antenna (MA) technologies to enhance the performance of SWIPT systems. The proposed alternating optimization approach is mathematically rigorous and demonstrates notable gains in harvested energy and communication reliability.
The main advantages of the work include:
- A novel integration of RIS and MA concepts that introduces additional spatial degrees of freedom for SWIPT systems.
- A mathematically solid optimization framework leveraging SCA and Taylor expansion for tractable performance maximization.
- Comprehensive simulation analysis demonstrating superior performance over RIS-only or fixed-antenna baselines.
However, the paper’s disadvantages and limitations should be addressed to improve completeness and practical value:
- The complexity and scalability of the proposed SCA-based algorithm are not analyzed. How does the approach scale with the number of RIS elements or users?
- The physical implementation feasibility of movable antennas (e.g., actuation latency, energy cost, mechanical constraints) is not discussed. Could the authors elaborate on how these issues might affect real-world deployment?
- The non-convex optimization structure is addressed with standard relaxations (SDR and SCA), but the convergence proof and sensitivity to initialization should be explicitly discussed.
- The comparison baseline could be expanded to include hybrid beamforming or meta-surface–assisted architectures for fairness.
To strengthen the theoretical and contextual foundation, it is suggested to include the following references, which provide both historical background and complementary approaches to beam synthesis and array optimization:
- Zhang, T., & Ser, W. (2011). Robust beampattern synthesis for antenna arrays with mutual coupling effect. IEEE Transactions on Antennas and Propagation, 59(8), 2889–2895. Relevant for discussing coupling-aware optimization in multi-antenna scenarios.
- Mohammed, J. R. (2024). Unconventional method for antenna array synthesizing based on ascending clustered rings. Progress In Electromagnetics Research Letters, 117, 69–73. Demonstrates innovative geometric array synthesis techniques, relevant for spatial adaptation discussions.
- Battaglia, G. M., Isernia, T., Palmeri, R., Maisto, M. A., Solimene, R., & Morabito, A. F. (2025). Near-field synthesis of 1-D shaped patterns through spectral factorization and minimally-redundant array-like representations. IEEE Transactions on Antennas and Propagation. Presents a deterministic framework that aligns conceptually with joint beamforming and RIS optimization.
- Taylor, T. T. (1955). Design of line-source antennas for narrow beamwidth and low side lobes. Transactions of the IRE Professional Group on Antennas and Propagation, 3(1), 16–28. A foundational reference in array synthesis, essential for contextual completeness.
- Orchard, H. J., Elliott, R. S., & Stern, G. J. (1985). Optimising the synthesis of shaped beam antenna patterns.IEE Proceedings H (Microwaves, Antennas and Propagation), 132(1), 63–68. Classical analytical treatment of shaped-beam optimization, forming the theoretical basis of modern beamforming.
- Battaglia, G. M., Isernia, T., Palmeri, R., & Morabito, A. F. (2023). Four-beams-reconfigurable circular-ring array antennas for monopulse radar applications. Radio Science, 58(9), 1–14. Demonstrates practical reconfigurable architectures with multi-beam control, offering conceptual links to the proposed MA-RIS integration.
Including these works will help position the current study within a broader historical and analytical context, reinforcing its originality and highlighting its practical implications in the evolution of reconfigurable and energy-efficient antenna systems.
Author Response
|
1. Summary: |
|
|
The paper presents a well-structured and technically sound framework that combines RIS and movable antenna (MA) technologies to enhance the performance of SWIPT systems. The proposed alternating optimization approach is mathematically rigorous and demonstrates notable gains in harvested energy and communication reliability. The main advantages of the work include: A novel integration of RIS and MA concepts that introduces additional spatial degrees of freedom for SWIPT systems. A mathematically solid optimization framework leveraging SCA and Taylor expansion for tractable performance maximization. Comprehensive simulation analysis demonstrating superior performance over RIS-only or fixed-antenna baselines.
Response: We sincerely thank you for the recognition of the strengths and contributions of our work. Your encouraging comments regarding the novelty of combining RIS and MA technologies, the mathematical rigor of our alternating optimization framework, and the comprehensiveness of our simulation validation are deeply appreciated. We are particularly grateful for your acknowledgment of how the introduced spatial degrees of freedom and the adopted approach contribute to performance enhancement in SWIPT systems. These insights strongly affirm the design motivations and technical direction of our study.
|
2. Point-by-point response to Comments and Suggestions for Authors
|
|
Comments 1: The complexity and scalability of the proposed SCA-based algorithm are not analyzed. How does the approach scale with the number of RIS elements or users?
Response 1: We sincerely thank you for raising this important point. To address this, we have provided a detailed analysis of the algorithm’s computational complexity and scalability in our responses to Reviewer #1 – Comment 1 and Reviewer #2, and also incorporated the complexity discussion into the revised manuscript. For detailed discussion, please refer to our previous response.
Comments 2: The physical implementation feasibility of movable antennas (e.g., actuation latency, energy cost, mechanical constraints) is not discussed. Could the authors elaborate on how these issues might affect real-world deployment?
Response 2: We thank you for raising this important and practical concern regarding the feasibility of MA. We fully agree that physical constraints such as actuation latency, mechanical precision, energy consumption, and reliability are crucial factors for real-world deployment of MA-assisted systems. While this work focuses on the algorithmic design and performance analysis under ideal actuation assumptions, we acknowledge that these hardware limitations may introduce delays, increase system complexity, and influence the achievable gains. To address this, we have added a brief but explicit discussion in the Conclusion section of the revised manuscript as a direction for future work. (Section 5) In the future, we will investigate the practical challenges associated with movable antenna implementations, including actuation latency, mechanical constraints, and energy overhead.
Thank you again for pointing out this valuable aspect.
Comments 3: The non-convex optimization structure is addressed with standard relaxations (SDR and SCA), but the convergence proof and sensitivity to initialization should be explicitly discussed.
Response 3: We thank you for pointing out this important aspect. The convergence of the proposed AO-based algorithm is guaranteed since each subproblem (i.e., P2, P3, and P4) is solved optimally or with guaranteed convergence using convex approximation techniques. The objective function is non-decreasing in each iteration and upper-bounded due to the practical transmission power constraints, thereby ensuring convergence. This has been clarified in the revised manuscript. (Section 3) In each iteration, one set of variables is optimized while fixing the others, and the non-convex subproblems are replaced by their tractable convex approximations. Since the objective function value is non-decreasing over iterations and the total transmit power is upper bounded, the convergence of the proposed algorithm is guaranteed. Moreover, as added in the revised manuscript, the proposed AO-SCA algorithm is initialized with random values in all simulations. Despite the non-convex nature of the problem, the algorithm demonstrates stable convergence behavior across different runs, showing low sensitivity to initialization and validating the robustness of the proposed approach. A clarification has been added to the manuscript accordingly. (Section 4) In our simulations, all variables are initialized randomly without requiring problem-specific tuning or fixed starting points.
Comments 4: The comparison baseline could be expanded to include hybrid beamforming or meta-surface–assisted architectures for fairness.
Response 4: We sincerely thank you for this constructive comment. In the current version of our manuscript, we have already included two widely-used baseline schemes for comparative analysis: (i) FPA-based RIS-SWIPT scheme, and (ii) a RIS system with randomly configured phase shifts. These benchmarks are representative of conventional non-movable RIS deployments and allow for fair evaluation of the impact brought by introducing MA. We agree that further expanding the comparison to include hybrid beamforming or other metasurface-assisted architectures could offer valuable insights. However, due to page limitations and the focus of this paper being the integration of MA into RIS-SWIPT systems, such comparisons are beyond the scope of this study. To address your suggestion, we have added a note in the conclusion section indicating that future work could explore the integration of the proposed framework with hybrid beamforming architectures and compare against intelligent metasurface-based systems with more advanced configurations. (Section 5) Moreover, we will consider integrating hybrid beamforming strategies into the proposed MA-RIS framework and conducting studies with other emerging metasurface-assisted architectures. We thank you again for this insightful suggestion, which will guide our extended research in follow-up studies.
Comments 5: To strengthen the theoretical and contextual foundation, it is suggested to include the following references, which provide both historical background and complementary approaches to beam synthesis and array optimization: [R1] Zhang, T., & Ser, W. (2011). Robust beampattern synthesis for antenna arrays with mutual coupling effect. IEEE Transactions on Antennas and Propagation, 59(8), 2889–2895. Relevant for discussing coupling-aware optimization in multi-antenna scenarios. [R2] Mohammed, J. R. (2024). Unconventional method for antenna array synthesizing based on ascending clustered rings. Progress In Electromagnetics Research Letters, 117, 69–73. Demonstrates innovative geometric array synthesis techniques, relevant for spatial adaptation discussions. [R3] Battaglia, G. M., Isernia, T., Palmeri, R., Maisto, M. A., Solimene, R., & Morabito, A. F. (2025). Near-field synthesis of 1-D shaped patterns through spectral factorization and minimally-redundant array-like representations. IEEE Transactions on Antennas and Propagation. Presents a deterministic framework that aligns conceptually with joint beamforming and RIS optimization. [R4] Taylor, T. T. (1955). Design of line-source antennas for narrow beamwidth and low side lobes. Transactions of the IRE Professional Group on Antennas and Propagation, 3(1), 16–28. A foundational reference in array synthesis, essential for contextual completeness. [R5] Orchard, H. J., Elliott, R. S., & Stern, G. J. (1985). Optimising the synthesis of shaped beam antenna patterns.IEE Proceedings H (Microwaves, Antennas and Propagation), 132(1), 63–68. Classical analytical treatment of shaped-beam optimization, forming the theoretical basis of modern beamforming. [R6] Battaglia, G. M., Isernia, T., Palmeri, R., & Morabito, A. F. (2023). Four-beams-reconfigurable circular-ring array antennas for monopulse radar applications. Radio Science, 58(9), 1–14. Demonstrates practical reconfigurable architectures with multi-beam control, offering conceptual links to the proposed MA-RIS integration. Including these works will help position the current study within a broader historical and analytical context, reinforcing its originality and highlighting its practical implications in the evolution of reconfigurable and energy-efficient antenna systems.
Response 5: We sincerely thank you for the helpful recommendation. We agree that incorporating foundational and recent works in beamforming and array synthesis will significantly enrich the context and theoretical underpinnings of our study. In the revised manuscript, we have cited the suggested references [R1]–[R6] and discussed their relevance in the Introduction section 1. Specifically, we added a paragraph to highlight the evolution of beamforming techniques and their essential role in supporting SWIPT systems. This addition reinforces the novelty and practical value of our proposed MA-RIS-assisted framework by situating it within the broader development of beam synthesis and reconfigurable architectures. (Section 1) Over the past decades, beam pattern synthesis has undergone development, providing both analytical foundations for emerging SWIPT systems. Classical works such as [R4] and [R5] for narrow-beam and sidelobe-controlled array structures. The practical implementation of SWIPT systems fundamentally relies on advanced beamforming and array synthesis techniques[R1-R3], [R6]. By shaping energy-efficient beams with spatial selectivity, beamforming enables targeted delivery of both information and energy, thereby serving as the backbone of SWIPT-enabled architectures. |
Thank you for the thorough evaluation and for highlighting the strengths of our work. Your valuable comments and suggestions have greatly helped us improve the clarity, completeness, and contextual relevance of the manuscript.
Best regards,
< All authors>

Reviewer 4 Report
Comments and Suggestions for Authors
- Keywords have to be sorted alphabetically.
- Authors are encouraged to give physical meanings in addition to equations in explaining why, for example, MA is chosen over RIS.
- The paper does not consider the limitations of MA.
- The study considered an ideal scenario; however, it should also include other scenarios where more channel parameters vary.
- The article has a strong mathematical foundation; however, many equations lack citations, and does this mean that the authors derived these equations? If not, please cite a reference to each equation that the authors do not derive.
- I suggest creating an appendix to relocate the derivations, provided this does not compromise the paper's clarity.
- The authors mentioned that they use simulations; however, they did not say what software they used.
- How complicated is it to set up the system and get results? Including computational cost.
- The authors considered a simulation only; how difficult is it to set a practical scenario?
- It may be beneficial to include an acronym list in the paper.
- In Figure 2, use different line types to be compatible with black-and-white printing.
- In Figure 5, define the unit of Pmax.
Author Response
|
1. Summary |
|
|
We sincerely appreciate your careful reading and constructive suggestions. Your comments have significantly helped us improve the clarity, rigor, and presentation quality of the manuscript. In response, we have:
- Reorganized the keywords in alphabetical order as suggested.
- Added physical insights to clarify the design motivation behind choosing Movable Antenna (MA).
- Discussed the ideal assumptions used in the system model and clarified that real-world hardware and channel variations are deferred to future work, with new content added in the conclusion accordingly.
- Moved complex derivations such as gradients and Hessian matrices to an appendix section to improve readability.
- Specified the simulation platform (MATLAB with CVX toolbox) and added discussions on computational complexity, setup difficulty, and deployment feasibility.
- Provided the unit for parameters such as Pmax, and adjusted figure styles (e.g., line types in Figure 2) to ensure print compatibility.
- Included an acronym list for better reader reference.
We appreciate your thoughtful suggestions, which have helped us improve both technical and editorial aspects of our work.
- 2. Point-by-point response to Comments and Suggestions for Authors
Comments 1: Keywords have to be sorted alphabetically.
Response 1: We appreciate your careful attention to detail. As suggested, we have reordered the keywords alphabetically in the revised manuscript to ensure compliance with editorial standards.
Internet of Things (IoT); Movable antenna (MA); Reconfigurable intelligent surfaces (RIS); Simultaneous wireless information and power transfer (SWIPT).
Comments 2: Authors are encouraged to give physical meanings in addition to equations in explaining why, for example, MA is chosen over RIS.
Response 2: We appreciate your comments. In the revised manuscript, we have included a more explicit physical explanation of the motivation for integrating MA in addition to RIS. While RIS enables passive beam manipulation via phase adjustments, its resolution may be limited due to the lack of adaptive physical repositioning. Movable antennas, on the other hand, contribute an additional spatial degree of freedom by allowing physical movement, which can compensate for imperfect RIS beam shaping and enhance the fine-grained focus of the energy and information beams. This synergy enables more precise beam steering and improved robustness in dynamic wireless environments. The physical interpretations have been added to both the introduction and system model sections.
(Section 1) When the reflected beam of the RIS lacks sufficient focus or alignment, the MA can provide fine-grained spatial adaptation to compensate for beamforming imperfections, thereby enhancing overall beam directivity and performance.
Comments 3: The paper does not consider the limitations of MA.
Response 3: Thank you for this insightful suggestion. In the revised manuscript, we have added a brief discussion regarding the practical limitations of MA deployment. Specifically, real-world MA implementations may face challenges such as actuation latency, mechanical constraints, energy consumption, and robustness in dynamic environments. These factors could impact the responsiveness and reliability of MA-assisted beamforming. However, addressing these issues requires hardware-oriented investigations and control-level integration, which are beyond the scope of the current theoretical study. We have accordingly stated this in the Conclusion section as an important direction for future work.
(Section 5) In the future, we will investigate the practical challenges associated with MA implementations, including actuation latency, mechanical constraints, and energy overhead.
Comments 4: The study considered an ideal scenario; however, it should also include other scenarios where more channel parameters vary.
Response 4: Thank you for your valuable comment. In this work, we consider perfect channel state information (CSI) and adopt a widely used field response channel model [23] to highlight the theoretical performance of the proposed MA-enhanced SWIPT system. We acknowledge that, due to the complexity and variability of wireless communication environments, incorporating diverse and more realistic channel scenarios (e.g., imperfect CSI, multipath fading, or channel uncertainty) is necessary for practical deployment. Our current study aims to provide a theoretical performance benchmark and a proof-of-concept reference. We will consider extending the analysis to more generalized channel conditions in our future work.
(Section 2) This paper aims to provide a theoretical performance benchmark and a proof-of-concept reference. We will consider extending the analysis to more generalized channel conditions in our future work.
Comments 5: The article has a strong mathematical foundation; however, many equations lack citations, and does this mean that the authors derived these equations? If not, please cite a reference to each equation that the authors do not derive.
Response 5: Thank you for this insightful remark. The majority of the equations presented in the manuscript are either derived by the authors or are direct results of standard formulations widely used in the literature. For clarity and academic rigor, we have carefully reviewed the manuscript and added appropriate citations for equations that are based on existing literature or widely accepted models (e.g., channel models, energy harvesting functions). Citations have now been included at their respective locations in the revised manuscript. All remaining equations without references are derived or are intermediate results based on the derived formulations. Thank you again for helping us improve the completeness and transparency of the paper. For example:
Furthermore, under a linear EH model [30], the amount of power harvested at the IoT node is given by
Comments 6: I suggest creating an appendix to relocate the derivations, provided this does not compromise the paper's clarity.
Response 6: Thank you for the valuable suggestion. To improve the readability and structural clarity of the manuscript, we have relocated the gradient and Hessian derivations to Appendix A. This reorganization preserves mathematical completeness while maintaining a clear narrative in the main text.
Comments 7: The authors mentioned that they use simulations; however, they did not say what software they used.
Response 7: Thank you for your comment. All simulations in this work were conducted using the MATLAB platform, specifically utilizing the CVX toolbox for convex optimization. The toolbox enables disciplined convex programming and is widely adopted in signal processing and wireless communication research. We have added this information in the revised manuscript for clarity.
(Section 4) All simulations were performed on MATLAB and the CVX toolbox.
Comments 8: How complicated is it to set up the system and get results? Including computational cost.
Response 8: Thank you for raising this practical question. In response to a similar and insightful comment by Reviewer #2, we have already added a detailed analysis of the computational complexity in the revised manuscript. Specifically, we provide an explicit expression for the complexity of each subproblem involved in the alternating optimization procedure and discuss how the algorithm scales with the number of RIS elements and users.
For clarity and completeness, please kindly refer to our response to Reviewer #2, and the corresponding revisions in the manuscript (Section 3.4.2), where the complexity expressions and implementation feasibility are thoroughly discussed.
Comments 9: The authors considered a simulation only; how difficult is it to set a practical scenario?
Response 9: Thank you for the insightful comment. Indeed, the current work is developed primarily from a theoretical and algorithmic perspective, aiming to evaluate the performance limits and optimization potential of integrating movable antennas (MAs) with reconfigurable intelligent surfaces (RIS) in SWIPT systems.
While building a fully functional prototype would require significant hardware efforts—such as real-time actuation control, energy harvesting circuitry, and robust channel estimation—this lies beyond the scope of our current study. Nevertheless, we emphasize that the proposed framework provides valuable design insights and performance benchmarks that can guide future system-level implementation.
To better reflect this, we have clarified in the conclusion that the current focus is on theoretical validation of the performance gains brought by MA–RIS joint design, while practical deployment challenges remain a promising direction for future research and engineering efforts.
(Section 5) This work focuses on the theoretical performance of MA–RIS-assisted SWIPT systems. In the future, we will investigate the practical challenges associated with MA implementations, including actuation latency, mechanical constraints, and energy overhead to bridge theory and implementation.
Comments 10: It may be beneficial to include an acronym list in the paper.
Response 10: We appreciate your thoughtful suggestion. We have now added a comprehensive list of acronyms used in the manuscript to enhance readability and accessibility for readers who may not be familiar with the technical terms.
Comments 11: In Figure 2, use different line types to be compatible with black-and-white printing.
Response 11: Thank you for the helpful suggestion. We have revised Figure 2 by incorporating different line types for each curve to ensure clarity and compatibility with black-and-white printing. The figure caption has also been updated accordingly.
Comments 12: In Figure 5, define the unit of Pmax.
Response 12: Thank you for pointing this out. We have revised the caption of Figure 5 to clearly indicate that the unit of Pmax is in dBm. This helps ensure clarity for the readers and aligns with standard practices in wireless communication simulations.
We sincerely appreciate your careful reading and insightful comments. Your constructive feedback has been instrumental in enhancing the presentation quality, clarity, and rigor of our manuscript. Thank you for your time and thoughtful contributions.
Best regards,
< All authors>

Round 2
Reviewer 1 Report
Comments and Suggestions for Authors
No more comment.
Reviewer 3 Report
Comments and Suggestions for Authors
The authors have satisfactorily addressed all the raised concerns. The paper is suitable for acceptance in its current form